# Self-Supervised Multi-Object Tracking with Cross-Input Consistency

**Favyen Bastani**
MIT CSAIL
favyen@csail.mit.edu

**Songtao He**
MIT CSAIL
songtao@csail.mit.edu

**Sam Madden**
MIT CSAIL
madden@csail.mit.edu

## Abstract

In this paper, we propose a self-supervised learning procedure for training a robust multi-object tracking (MOT) model given only unlabeled video. While several self-supervisory learning signals have been proposed in prior work on single-object tracking, such as color propagation and cycle-consistency, these signals cannot be directly applied for training RNN models, which are needed to achieve accurate MOT: they yield degenerate models that, for instance, always match new detections to tracks with the closest initial detections. We propose a novel self-supervisory signal that we call cross-input consistency: we construct two distinct inputs for the same sequence of video, by hiding different information about the sequence in each input. We then compute tracks in that sequence by applying an RNN model independently on each input, and train the model to produce consistent tracks across the two inputs. We evaluate our unsupervised method on MOT17 and KITTI — remarkably, we find that, despite training only on unlabeled video, our unsupervised approach *outperforms* four supervised methods published in the last 1–2 years, including Tracktor++ [1], FAMNet [5], GSM [18], and mmMOT [29].

## 1   Introduction

Multi-object trackers identify all instances of a particular object type in video, and track each instance through the segment of video in which it is visible in the camera frame. Annotating training data for multi-object tracking is tedious and costly; for example, annotation of pedestrian tracks in just six minutes of video in the training set of the MOT15 Challenge [14] requires an estimated 22 hours [20] of human labeling time using LabelMe [28]. While unsupervised, heuristic detect-to-track methods [2, 4] have been proposed that group detections into tracks by estimating motion using a combination of spatial and visual cues, these methods suffer low-accuracy in scenarios with frequent occlusion where heuristics are insufficient.

Recent work has proposed applying self-supervised learning for training *single*-object tracking models on unlabeled video [24, 25]. These approaches train a model to propagate instance labels from a reference frame through the rest of a video sequence. In contrast to work on self-supervised representation learning from video, these fully unsupervised approaches do not require fine-tuning to apply the model for single-object tracking.

However, a significant limitation in prior work is that the model independently compares pairs of frames at a time. In multi-object tracking, a key challenge is robustly re-localizing tracks across potentially long occlusions, especially when an object instance is occluded by other instances of the same object type. Pairwise frame comparisons are thus insufficient for high-accuracy multi-object tracking; instead, learning recurrent features that encode the history of a track is crucial for enabling robust re-localization. However, extending prior work to learn RNN parameters is challenging. For example, Wang et al. [25] propose training using forward-backward consistency: from a patch in an initial frame, after tracking forwards through video and then backwards to return to the initial frame,

the final patch should align with the original patch. Training an RNN in this way would be ineffective as the RNN could simply memorize the features of the original patch.

To address this challenge, we propose a novel self-supervised learning method, *cross-input consistency*. We first compute object detections in each frame of unlabeled video (like unsupervised, heuristic detect-to-track methods, we assume that a robust detector is available). Then, we derive a learning signal from the unlabeled video by sampling a short sequence of contiguous frames from the video, constructing two input variations of that sequence that each hide different information about objects detected in the sequence, and training the tracker to produce consistent tracking outputs when applied independently on each of the two inputs. We propose two alternative input-hiding schemes for computing the input variations: visual-spatial hiding and occlusion-based hiding. Visual-spatial hiding applies the tracker once when only observing spatial inputs (bounding box coordinates in the video frame), and once when only observing visual inputs (pixel values inside detection boxes). Occlusion-based hiding eliminates information about object detections in random intermediate subsequences of frames to simulate occlusion incidents; thus, it constructs two inputs by eliminating different subsequences of detections in each input. After sampling a sequence of video and computing the two input variations under the chosen input-hiding scheme, we apply the tracker model independently on each input, and back-propagate a learning signal that measures the consistency between tracks computed across the two inputs. To attain high consistency, the model must accurately group detections that correspond to the same object: if the model were to instead arbitrarily group detections into tracks, then variations in the inputs would cause the tracker to produce inconsistent outputs.

To implement cross-input consistency, we adapt a now standard RNN model and tracker architecture from prior work [12]: the tracker processes each frame in sequence by matching detections in the current frame with tracks computed up to the previous frame. In prior work, this model is trained under a supervised procedure: they sample a video sequence $\langle I_0, \ldots, I_n \rangle$ and a track $t$ in that sequence, and apply the tracker on $t$ over the sequence. On each frame $I_j$, the RNN outputs a probability distribution indicating the likelihood that the prefix of a track $t$ up to $I_j$ matches with each detection in $I_j$. Prior work back-propagates the label (i.e., the correct detection of $t$ in $I_j$) under cross entropy loss.

In contrast, under our method, on each training iteration, we propose to sample a sequence $\langle I_0, \ldots, I_n \rangle$ from a corpus of unlabeled video, and apply the RNN model to compute a transition matrix that specifies the probability that each detection in $I_0$ (rows) matches with each detection in $I_n$ (columns). We select the sequence length $n$ so that most objects in $I_0$ are still visible in $I_n$. Then, when applying the tracker on two input variations extracted from the sequence, we obtain two transition matrices (one for each input). We compute the dot-product similarity to measure the consistency between these matrices, and back-propagate the negative similarity as a loss function.

We evaluate our approach on the MOT17 and KITTI benchmarks against 9 baselines, including both unsupervised and supervised methods. We train our tracker model using cross-input consistency over a corpus of unlabeled video, which can be cheaply obtained. Like other unsupervised methods, we use an object detector trained on image-level bounding box annotations in COCO [17], but do not use any expensive video-level annotations. We find that, on MOT17, our approach improves both IDF1 and MOTA accuracy over the unsupervised baselines by 14% to 18%. Moreover, remarkably, our fully unsupervised approach *outperforms* five of the seven supervised methods we compared, even though these methods train on expensive video-level bounding box and track annotations.

Our code is available at `https://favyen.com/uns20/`.

## 2  Related Work

Self-supervised learning over video has been studied extensively in many contexts. Most work focuses on learning representations of video that can be applied through fine-tuning for tasks such as activity recognition, image classification, and object detection [6, 7, 9, 15, 23, 26]. More closely related to our work, several recent approaches have proposed leveraging widely available unlabeled video to directly train *single*-object tracking models, without needing fine-tuning [13, 16]. Vondrick et al. [24] train a model to colorize gray-scale video by propagating colors from a colored reference frame. The model is then applied to track objects at inference time by propagating instance IDs instead of colors. Wang et al. [25] train a model to capture correspondence by applying a cycle-consistent loss: from

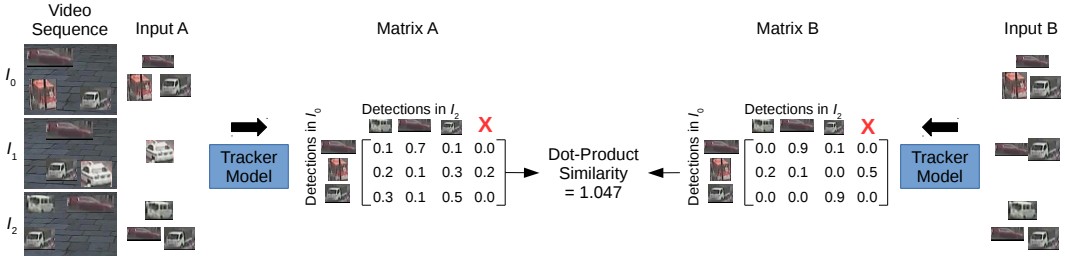

Figure 1: Overview of cross-input consistency. An input-hiding scheme produces two inputs, A and B, from one video sequence; these inputs contain identical information about objects detected in the first and last frames of the sequence, but vary in intermediate frames. We apply the tracker model on each input to derive two transition matrices that match detections between the first and last frames to represent tracker outputs. We then back-propagate a similarity score between the matrices that encourages the model to produce consistent outputs across both inputs.

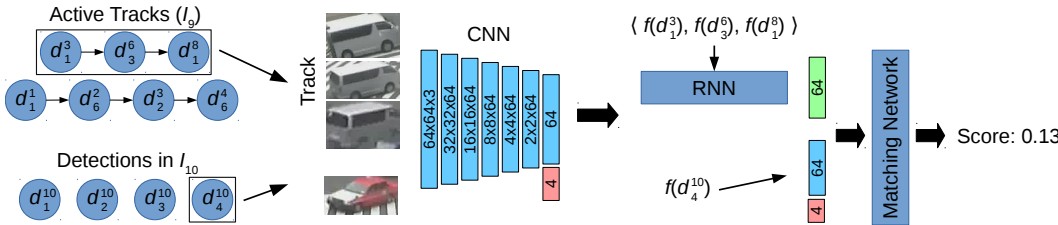

Figure 2: Tracker model architecture. The model scores the likelihood that each detection in the current frame matches with each track.

a patch in an initial frame, after tracking forwards through video and then backwards to return to the initial frame, the final patch should align with the original patch. As we discussed in Section 1, self-supervisory signals used in prior work such as color propagation and cycle-consistency are not effective for training RNN models, which are needed to achieve accurate MOT.

Our work is also related to unsupervised, heuristic detect-to-track multi-object tracking methods such as SORT [2] and V-IOU [4]. These methods group detections across different frames using a combination of heuristic spatial cues (e.g., Kalman filter over bounding box coordinates) and visual cues (e.g., optical flow) to track objects. Like our approach, these methods assume that a robust detector is available; however, because they rely on heuristics to group detections into tracks, they suffer low-accuracy in scenarios with frequent occlusion.

Multi-object tracking has been studied extensively in supervised settings, where methods are trained on video-level bounding box and track annotations [1, 5, 8, 12, 18, 30]. However, such annotations are expensive to hand-label, and so these methods are costly to extend to new types of video.

Other work explores using unsupervised and self-supervised learning to further improve the performance of fully supervised methods. SimpleReID [11] proposes improving the performance of one supervised method, CenterTrack [30], by training a re-identification model through unsupervised learning. However, while the model can in principle be trained only on image-level annotations through hallucinated motion techniques, their SimpleReID+CenterTrack tracking method depends on expensive video-level annotations to attain high-accuracy. In contrast, our method achieves competitive results without any video-level supervision.

## 3   Cross-Input Consistency

In our novel cross-input consistency method, we derive an learning signal for training an RNN tracker model through a three-step process. We assume that a corpus of unlabeled video is available, along with an object detector for the object category of interest. During pre-processing, we apply the detector on each frame of unlabeled video to compute object detections. Then, during training,

we first repeatedly randomly sample a sequence of contiguous frames from the video, $\langle I_0, \ldots, I_n \rangle$, where each $I_k$ is a video frame. Let $D_k$ by the detections automatically computed in $I_k$ by the detector, and let $d_i^k = (im, x, y, w, h)$ be a detection in $D_k$, where $(x, y, w, h)$ are the 4D spatial coordinates (center point and lengths) of the detection bounding box, and $im$ is the window of $I_k$ corresponding to that box. We apply an input-hiding scheme to select two input variations $A(D), B(D)$ for the video segment, where each variation is a modified sequence of detections in the frames, $A(D) = \langle D_0^A, \ldots, D_n^A \rangle$, $B(D) = \langle D_0^B, \ldots, D_n^B \rangle$. For example, some detections may be removed entirely, while others may be partially hidden. Second, we apply the tracker model independently on each input variation to derive two probabilistic tracking outputs (one per input), represented as transition matrices. Third, we compare the transition matrices with dot-product similarity to update the RNN parameters.

Figure 1 summarizes our approach.

Below, we first introduce the model architecture that we adapt from prior work in Section 3.1. We then detail our novel training procedure, including the computation of the transition matrices and dot-product loss, in Section 3.2. Finally, we propose two input-hiding schemes for selecting the input variations required by our approach in Section 4.

## 3.1 Background: Tracker Model

We adopt a tracker model that is similar to prior work [12]. We summarize the architecture in Figure 2. Given a video sequence $\langle I_0, \ldots, I_n \rangle$, and sets of detections $D_k = \{d_1^k, \ldots, d_{m_k}^k\}$ detected in each frame $I_k$, to initialize the tracking process, we create a length-1 track $t_i = \langle d_i^0 \rangle$ for each detection $d_i^0$ in the first video frame $I_0$. When processing subsequent frames, we will match the new detections with existing tracks, extending existing tracks if there is a match and initializing new tracks otherwise. Specifically, on each subsequent frame $I_k$, the model outputs a probability $p_{i,j}$ that each track $t_i$ corresponds to each detection $d_j^k \in D_k$. At inference time, we formulate the problem of matching tracks with detections in $I_k$ as a bipartite matching problem, where the cost of matching $t_i$ with $d_j^k$ is $1 - p_{i,j}$. We solve this problem and compute a minimum-cost matching using the Hungarian algorithm; for each pair $(t_i, d_j^k)$ in the matching, we append $d_j^k$ to $t_i$. For each detection in $I_k$ that no track matches to, we create a new track for that detection.

The model consists of a CNN, RNN, and matcher network. Together, these components score the likelihood that the $i$th track, $t_i = \langle d_1, \ldots, d_m \rangle$, matches with the $j$th detection in $I_k$, $d_j^k$. We first apply the CNN to derive detection-level features. Given a detection $d = (im, x, y, w, h)$, the CNN inputs $im$ resized to $64 \times 64$, and consists of 6 strided convolutional layers, with ReLU activation in the first 5 layers and linear activation in the last layer. It outputs a 64-vector, which we concatenate with the 4D spatial coordinates to derive a 68-vector detection representation $f(d)$. Then, we compute track-level features $f(t_i)$ by applying the RNN (an LSTM with 64 hidden states) over the sequence of detection-level features of detections in the track, $\langle f(d_1), \ldots, f(d_m) \rangle$. We use the output of the RNN on the last timestep as the track-level features $f(t_i)$. Finally, we apply a matching network to score the likelihood that $t_i$ matches $d_j^k$. The matching network inputs the concatenation of $f(t_i)$ and $f(d_j^k)$, applies four fully-connected layers, and outputs a match score.

## 3.2 Training Procedure

We develop a novel self-supervised learning method for training the model parameters on unlabeled video. During training, we repeatedly sample sequences of video $\langle I_0, \ldots, I_n \rangle$. We apply one of two input-hiding schemes, which we will detail in the following section, to extract two distinct input variations $A(D)$ and $B(D)$ from a sampled video sequence, where each input is a sequence of detections. We then apply the tracker independently on $A(D)$ and $B(D)$ to derive two tracking outputs for the same video sequence. In cross-input consistency, we train the model by enforcing similarity between these two outputs.

To represent tracker outputs, we compute an $|D_0| \times |D_n| + 1$ transition matrix $M^{(0,n)}$, where $M_{i,j}^{(0,n)}, j < |D_n|$ is the probability that the track $t_i$ matches $d_j^n$. We use the last column to represent tracks that are no longer visible in $I_n$, i.e., $M_{i,|D_n|}^{(0,n)}$ is the probability that the track $t_i$ has exited the camera frame. When applying the model over video sequences during training, we update tracks with

new detections based on the scores output by the model on intermediate frames, but do not create additional tracks on frames after $I_0$; thus, each track $t_i$ corresponds directly to a detection $d_i^0$ in $I_0$ (i.e., $t_i = \langle d_i^0, \ldots \rangle$). Thus, we can also think of $M^{(0,n)}$ as the probability that a detection in the first frame $d_i^0$ matches a detection in the last frame $d_j^n$.

Applying the tracker on both input variations yields two transition matrices $A^{(0,n)}$ and $B^{(0,n)}$ that match objects detected in $I_0$ with those in $I_n$. We train the model (CNN, RNN, and matching network) to maximize the dot-product similarity between these matrices. In addition to pushing the model to produce consistent outputs across both inputs, we also design our training method so that the model cannot attain a high similarity score by, for example, saying that all objects visible in $I_0$ are no longer visible in $I_n$.

In our method, it is important that the training sequence length $n$ be chosen so that, in most sequences, most (but not all) objects in $I_0$ are still visible in $I_n$, but that objects nevertheless move non-trivially during the sequence (so that the tracking task is not too easy). In general, we find that setting $n$ to one-half of the average time that objects linger in the camera frame works well; this value can be quickly estimated by hand-labeling the duration of a few (e.g., 10-20) objects randomly sampled from the video.

Below, we detail our method to compute transition matrices, and discuss dot-product similarity loss.

**Transition Matrix.** We propose computing a transition matrix $M^{(0,k)}$ on each frame $I_k$ to represent the tracker outputs, where $M_{i,j}^{(0,k)}$ is the probability that the track $t_i$ matches the detection $d_j^k$. On intermediate frames, we apply the Hungarian method on $M^{(0,k)}$ to match detections in $D_k$ with tracks, updating each track with the matched detection (if any). On the last frame $I_n$, we use the $M^{(0,n)}$ matrix produced under different inputs (denoted $A^{(0,n)}$ and $B^{(0,n)}$) to compute and back-propagate a consistency score. Because we do not create new tracks after $I_0$ during training, $M_{i,j}^{(0,n)}$ is the likelihood that $d_i^0$ and $d_j^n$ match (since $t_i$ begins with $d_i^0$).

We first construct a score matrix $S^{(0,k)}$, by computing $S_{i,j}^{(0,k)}$ as the score (any real number) output by the tracker model given the track $t_i$ and detection $d_j^k$. We then transform the score matrix into a probability matrix to derive $M^{(0,k)}$. We could simply compute $M^{(0,k)}$ by taking softmax along rows in $S^{(0,k)}$. However, computing the transition matrix in this way would allow the tracker to cheat and maximize similarity between $A^{(0,n)}$ and $B^{(0,n)}$ by simply matching all detections in $I_0$ to a single detection $d_j^n \in D_n$. Indeed, we find that in practice this yields degenerate models.

Thus, instead, we compute $M^{(0,k),\text{row}}$ and $M^{(0,k),\text{col}}$ by applying softmax along rows and columns, respectively, and compute $M^{(0,k)} = \min(M^{(0,k),\text{row}}, M^{(0,k),\text{col}})$:

$$M_{i,j}^{\text{row}} = \frac{\exp(S_{i,j})}{\sum_k \exp(S_{i,k})} \qquad M_{i,j}^{\text{col}} = \frac{\exp(S_{i,j})}{\sum_k \exp(S_{k,j})} \qquad M_{i,j} = \min(M_{i,j}^{\text{row}}, M_{i,j}^{\text{col}}) \quad (1)$$

This produces a transition matrix $M^{(0,k)}$ that is almost doubly stochastic: rows and columns sum to at most 1, but not necessarily exactly 1. The operation ensures that the model must match each detection in $I_0$ to unique detections in $I_n$ to maximize the consistency score between $A^{(0,n)}$ and $B^{(0,n)}$: if two detections in $I_0$ are matched to the same detection in $I_n$, then the columnar softmax would reduce those probabilities in the corresponding matrix to at most 0.5, thereby reducing any dot-products involving those rows.

**Dot-Product Similarity.** We train the RNN tracker by computing two transition matrices $A^{(0,n)}$ and $B^{(0,n)}$ over different input variations, and then back-propagating a loss that measures the inconsistency between the matrices. In particular, we use the dot-product to measure the similarity of corresponding rows in the matrices. We define the loss as:

$$L = -\sum_i \log \sum_j A_{i,j}^{(0,n)} B_{i,j}^{(0,n)}$$

Here, $L$ is computed by taking the logarithm of the dot product of corresponding rows in $A^{(0,n)}$ and $B^{(0,n)}$, averaged across rows. Note that this is equivalent to the cross-entropy loss between the diagonal matrix and the matrix product of $A^{(0,n)}$ and the transpose of $B^{(0,n)}$.

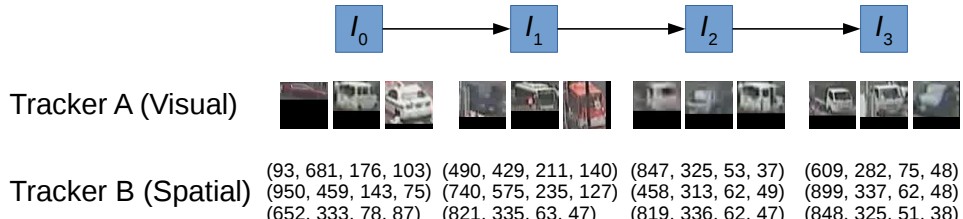

Tracker A (Visual)

Tracker B (Spatial)

(93, 681, 176, 103) (490, 429, 211, 140) (847, 325, 53, 37) (609, 282, 75, 48)
(950, 459, 143, 75) (740, 575, 235, 127) (458, 313, 62, 49) (899, 337, 62, 48)
(652, 333, 78, 87) (821, 335, 63, 47) (819, 336, 62, 47) (848, 325, 51, 38)

Figure 3: Visual-spatial hiding. One input includes only visual information about the detections, while the other includes only spatial bounding box coordinates.

This loss function has several desirable properties. First, the dot-product is maximized when the matrices computed based on different inputs are most similar. This pushes the matching model to learn reasonable visual and spatial tracking constraints, because arbitrarily matching tracks to detections will lead to dissimilarity. Second, the dot-product pushes each matrix to be almost doubly stochastic rather than leaving some rows and columns summing to much less than 1. The model can only produce doubly stochastic $A^{(0,n)}$ and $B^{(0,n)}$ matrices by finding unique detections in $I_n$ for each detection in $I_0$.

**Spatial Mask.** In some cases, training as described above may converge at a local minimum where the model outputs uniform probabilities for all entries in $M$. To mitigate this issue, we add a spatial constraint to the loss that penalizes the tracker when it matches detections that are highly improbable to correspond to the same object based on bounding box positions. We first compute a mask matrix $C^{(0,n)}$ so that $C_{i,j}^{(0,n)} = 0$ if it is "improbable" that $d_i^0$ matches $d_j^n$, and $C_{i,j}^{(0,n)} = 1$ otherwise. Then, we compute $L$ as:

$$L = -\sum_i \log \sum_j A_{i,j}^{(0,n)} B_{i,j}^{(0,n)} C_{i,j}^{(0,n)}$$

We determine whether matches in $C$ are improbable by applying a simple floodfill-like algorithm that propagates sets of labels from the first frame $I_0$ to the last frame $I_n$. If the label from a detection $d_i^0$ in $I_0$ does not propagate to a detection $d_j^k$, then it implies there is no sequence of intermediate detections that could form a track between $d_i^0$ and $d_j^k$. In $I_0$, we label each detection $d_i^0$ with a set containing only that detection, i.e., $\{d_i^0\}$. In $I_k$, we label each detection $d_j^k$ with the union of sets of labels of detections $d_i^{k-l}$ in preceding frames $I_{k-l}$ ($1 \leq l \leq 10$) such that the bounding boxes of $d_j^k$ and $d_i^{k-l}$ intersect. Note that we consider several preceding frames since the detector may occasionally fail to localize an object in an intermediate frame. Then, $C_{i,j}^{(0,n)} = 1$ only if the label set for $d_j^n$ includes $d_i^0$.

**Artificial Detections.** To improve the model's robustness in learning visual features, we artificially construct additional detections in $I_n$ by pairing the spatial coordinates of detections in $I_n$ with object images selected randomly from frames in the underlying video data that are temporally far from $\langle I_0, \ldots, I_n \rangle$. Thus, these artificial detections added to $D_n$ have correct spatial coordinates, but include visual cues that do not correspond to any object in $I_0$, and so the tracker model must learn to leverage visual cues so that it does not assign high probabilities in $M^{(0,n)}$ to artificial detections.

We exclude artificial detections in the mask $C$. Then, to perform well under dot-product similarity, the model must learn to leverage visual features to distinguish the correct detections in $I_n$ from artificially constructed ones — a tracker that only considers spatial features would assign half of its probability mass along each row to artificial detections, and thus would be penalized by the loss.

## 4 Input-Hiding Schemes

In this section, we detail two alternative input-hiding schemes for selecting the two input variations, denoted $A(D) = \langle D_0^A, \ldots, D_n^A \rangle$ and $B(D) = \langle D_0^B, \ldots, D_n^B \rangle$. Recall that $D$ is the original set of all objects detected in a video sequence $\langle I_0, \ldots, I_n \rangle$. Although we only introduce two schemes, our cross-input consistency framework is general-purpose, and there may be other input-hiding schemes that offer comparable or better performance.

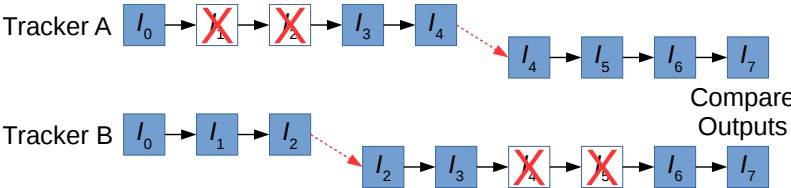

Figure 4: Occlusion-based hiding produces input variations with different subsequences of occluded frames where all detections are hidden from the tracker. It also independently applies the tracker before and after a hand-off frame ($I_4$ and $I_2$), and merges the outputs through the matrix product.

## 4.1 Visual-Spatial Hiding

Under visual-spatial hiding, we apply one tracker instance that observes only visual features and one tracker instance that observes only spatial features: in $A(D)$, we set $x = 0, y = 0, w = 0, h = 0$ for all detections (hide all spatial information), and in $B(D)$, we set $im = 0$ (hide the image content).

Training with cross-input consistency forces the model to produce similar outputs between the visual and spatial inputs. We find that, in practice, the model naturally learns to robustly track objects, because doing anything else would not lead to high consistency. For example, when given visual features, the model must learn to match detections based on visual similarity in order to be consistent with matching based on spatial proximity. Similarly, when given spatial features, the model must learn to estimate motion across occlusion, since the visual-only instance would not have difficulty re-localizing a visually distinctive object following frames where it was occluded.

A key issue with visual-spatial hiding is that the visual-only instance can estimate the change in spatial coordinates of an object between two frames by comparing the background of the detection bounding boxes of that object across the frames, similar to optical flow. This reduces performance because the visual-only instance then learns only to process the background rather than learn a robust embedding for contrasting distinct objects. To mitigate the issue, we make adjustments to the training and inference procedures.

**Training.** We prevent the visual-only instance from focusing on background features when matching detections in $I_n$ with those in $I_0$ by encumbering its ability to aggregate estimates of changes in spatial coordinates across sequences of consecutive frames: although processing the background is sufficient to compute spatial movement between detections that are in close proximity (e.g., detections of the same object between consecutive frames), this strategy fails for detections that share no overlap. Thus, for sufficiently large $n$, where objects move substantially during the sampled video sequence, background processing is only an issue because the tracker can add up changes that it computes between each pair of consecutive frames.

In particular, we eliminate the recurrent unit for the visual-only instance: its matching network inputs visual features for detections in $I_0$ and for detections in $I_n$, and scores each pair of detections in $A^{(0,n)}$ without observing intermediate frames. On the other hand, we make no changes to the spatial-only instance: it computes $B^{(0,n)}$ by processing 4D spatial coordinates for each detection in every frame in the sequence, and employs both the recurrent unit and the matching network. Figure 3 illustrates the training procedure.

**Inference.** The separation of visual and spatial inputs, and the specialized training procedure that we employ, imposes two challenges during inference. First, because the visual-only and spatial-only instances observe very different inputs, we cannot expect the model to perform well when we provide both inputs—in effect, we have trained two separate models. Second, since we eliminated recurrent features for the visual-only instance during training, the visual-only instance is essentially a re-identification model, and we must decide how to apply it during inference to take advantage of multiple prior observations of a track in previous frames.

To address the first challenge, during inference, for an input video sequence $\langle I_0, \ldots, I_k \rangle$, we independently compute the visual-only tracker output $A^{(0,k)}$ and the spatial-only tracker output $B^{(0,k)}$. We then compute $M^{(0,k)}$ as the minimum of $A^{(0,k)}$ and $B^{(0,k)}$, and use $M^{(0,k)}$ to update tracks before processing the next frame. Taking the minimum for a matching between track $t_i$ and detection $d_j^k$

ensures that the final transition probability reflects whichever of the visual features or spatial features that make $d_j^k$ less likely to match with $t_i$. This is desirable—for example, two red sedans visible in the same segment of video will have high visual similarity, and we may have to leverage spatial features to distinguish them.

The second challenge is that the visual-only tracker lacks recurrent features, and thus is only able to compare pairs of frames. To address this, when using the visual-only tracker, for each track-detection pair $(t_i, d_j^k)$, we compute 5 scores by applying the model on $d_j^k$ and 5 randomly selected images associated with the track $t_i$ in previous frames. We then average these scores to derive $A_{i,j}^{(0,k)}$. This enables the model to use context from multiple preceding frames when localizing an object in a new frame.

### 4.2 Occlusion-based Hiding

We also experimented with an occlusion-based hiding scheme. Since we found that visual-spatial hiding performs better, we introduce occlusion-based hiding only at a high level here, but include details in the supplementary material.

Occlusion-based hiding produces the variations $A(D)$ and $B(D)$ by simulating random occlusion incidents where all detections in occluded frames are eliminated from the input, i.e., if $I_k$ is occluded for $A(D)$, then $D_k^A$ is empty. We only occlude intermediate frames (i.e., a frame $I_k$ is only considered for occlusion if $0 < k < n$) so that the transition matrices still compare detections in $I_0$ with those in $I_n$. When processing an occluded $I_k$, the tracker is forced to match all tracks to the absent column in that frame, and re-localize the tracks after the occlusion. In occlusion-based hiding, we also incorporate an RNN hand-off method that cuts off the propagation of RNN features from $I_0$ to $I_n$ by employing two separate RNN executions: for some handoff index $1 < h < n$, we apply the model from $I_0$ to $I_h$, and separately apply the model from $I_h$ to $I_n$, and combine the transition matrices by taking their product. We summarize the scheme in Figure 4.

## 5 Evaluation

We compare our method and nine baselines on the MOT17 [21] and KITTI [10] benchmarks.

**Baselines.** We compare with two unsupervised methods (SORT [2] and V-IOU [4]) and seven fully supervised methods (Tracktor++ [1], MHT-BLSTM [12], FAMNet [5], LSST [8], GSM [18], mmMOT [29], and CenterTrack [30]). Like our approach, SORT and V-IOU require an object detector, but do not train on any video-level bounding box and track annotations in the MOT17 and KITTI training sets. The fully supervised methods train on video-level annotations; Tracktor++ incorporates a core component that uses only the detector regression network, but requires video-level annotations for training a re-identification network. Results for 8 baselines are available on MOT17, and results for 4 baselines are available on KITTI.

**Dataset.** MOT17 [21] consists of 14 video sequences of pedestrians in a wide range of contexts, including a moving camera inside a shopping mall and a fixed, elevated view of an outdoor plaza. The dataset is split into 7 training sequences and 7 test sequences; each split includes approximately 11 minutes of video. KITTI [10] consists of 48 video sequences captured from vehicle-mounted cameras, split into 20 for training and 28 for testing, and the objective is to track cars.

**Training.** The supervised baselines train on video-level bounding box and track annotations provided by MOT17 and KITTI. In contrast, our method trains only on a corpus of unlabeled video. Because video-level annotations are expensive to label, our method requires substantially less annotation time, and thus greatly reduces the effort needed to apply multi-object tracking on new datasets.

For MOT17, we collect unlabeled video from two sources: we use five hours of video from seven YouTube walking tours, and all train and test sequences from the PathTrack dataset [20] (we do not use the PathTrack ground truth annotations). For KITTI, we use both the 46 minutes of video in the KITTI dataset together with 7 hours of video from Berkeley DeepDrive [27]. We train our tracker model on an NVIDIA Tesla V100 GPU; training time varies between 4 and 24 hours depending on the input-hiding scheme. During training, we randomly select sequence lengths $n$ between 4 and 16

| Method | IDF1 | MOTA |
|---|---|---|
| Occlusion (ours) | 52.4 | 56.7 |
| Visual-Spatial (ours) | **57.3** | **60.2** |
| Spatial-Only (ours) | 56.5 | 57.8 |

Table 1: Ablation study on the MOT17 training set.

| | Method | **IDF1** | **MOTA** | MT | ML | FP | FN | Idsw | Frag |
|---|---|---|---|---|---|---|---|---|---|
| Unsupervised Methods | Visual-Spatial (ours) | **58.3** | **56.8** | 538 | 880 | 12K | 231K | 1K | 2K |
| | SORT [2] | 39.8 | 43.1 | 295 | 997 | 28K | 288K | 5K | 7K |
| | IOU [3] | 39.4 | 45.5 | 369 | 953 | 20K | 282K | 6K | 7K |
| | Tracktor++ [1] | 52.3 | 53.5 | 459 | 861 | 12K | 248K | 2K | 5K |
| | MHT-BLSTM [12] | 51.9 | 47.5 | 429 | 981 | 26K | 268K | 2K | 3K |
| Supervised Methods | FAMNet [5] | 48.7 | 52.0 | 450 | 787 | 14K | 254K | 3K | 5K |
| | LSST [8] | 62.3 | 54.7 | 480 | 944 | 26K | 228K | 1K | 4K |
| | GSM [18] | 57.8 | 56.4 | 523 | 813 | 14K | 230K | 1K | 3K |
| | CenterTrack [30] | 59.6 | 61.5 | 621 | 752 | 14K | 201K | 3K | 5K |

Table 2: Performance on the MOT17 test set. We compare methods in terms of IDF1 and MOTA, but include other non-comprehensive metrics from MOT17 as well for completeness.

| Method | **HOTA** | DetA | AssA | DetRe | DetPr | AssRe | AssPr | LocA |
|---|---|---|---|---|---|---|---|---|
| Visual-Spatial (ours) | **62.5** | 61.1 | 65.3 | 67.7 | 73.8 | 69.1 | 83.1 | 80.3 |
| SORT [2] | 42.5 | 44.0 | 41.3 | 47.3 | 73.9 | 42.8 | 83.0 | 80.8 |
| FAMNet [5] | 52.6 | 61.0 | 45.5 | 64.4 | 78.7 | 48.7 | 77.4 | 81.5 |
| mmMOT [29] | 62.1 | 72.3 | 54.0 | 76.2 | 84.9 | 59.0 | 82.4 | 86.6 |
| CenterTrack [30] | 73.0 | 75.6 | 71.2 | 80.1 | 84.6 | 73.8 | 89.0 | 86.5 |

Table 3: Performance on the KITTI test set (tracking cars). We show unsupervised methods, including our approach, at the top, and methods that require video-level annotations at the bottom. We use HOTA to compare methods, but include other non-comprehensive metrics from KITTI for completeness.

frames, and apply stochastic gradient descent one sequence at a time. We apply the Adam optimizer with learning rate 0.0001, decaying to 0.00001 after plateau.

In contrast to MOT17, KITTI does not provide object detections for use by tracking methods. We extract detections from video using a YOLOv5 model trained on COCO. On MOT17, we pre-process the provided Deformable Parts Model, Faster R-CNN, and Scale-Dependent Pooling detections with classification and regression following the pre-processing method in Tracktor++ [1].

**Metrics.** We use Multi-Object Tracking Accuracy (MOTA) [21] and ID F1 Score (IDF1) [22] on MOT17, and Higher Order Tracking Accuracy (HOTA) [19] for KITTI. Broadly, these comprehensive metrics measure the accuracy of inferred tracks against ground truth tracks, and penalize both when an inferred track contains a detection that doesn't match to some ground truth detection (or vice versa), and when a ground truth track is split into two or more inferred tracks (or vice versa). MOT17 and KITTI employ several other non-comprehensive metrics, many of which are used to compute MOTA, IDF1, and HOTA; we report these for completeness.

**Ablation Study.** We first compare occlusion-based hiding and visual-spatial hiding on the MOT17 training set in Table 1. Visual-spatial hiding yields higher performance on both MOTA and IDF1 — because objects are often visible in the video for only a short duration, when training under occlusion-based hiding, the model is unable to learn to re-localize objects over simulated occlusions since the simulated occlusion must then also be short. Under Spatial-Only, we show results for visual-spatial hiding when inputting only the spatial coordinates of detections during inference (no image features).

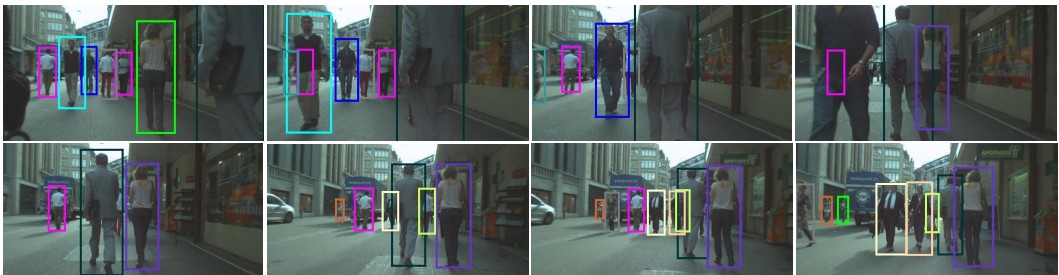

Figure 5: Output of Visual-Spatial on a portion of an MOT17 sequence. Our method tracks objects through several instances of occlusion.

**Quantitative Results.** Table 2 shows results on the MOT17 test set[1], and Table 3 shows results on the KITTI test set[2]. Metrics are automatically computed by the challenge websites. Per the challenge policies, we only submit the best method, and thus show Visual-Spatial performance.

On MOT17, our approach substantially outperforms both of the unsupervised baselines. Moreover, despite training only on unlabeled video, our method *outperforms* Tracktor++ [1], MHT-BLSTM [12], FAMNet [5], and GSM [18], even though these baselines (all of which except MHT-BLSTM were published in the last 1–2 years) are supervised methods that train on expensive video-level annotations in the MOT17 training set. Our approach is also competitive with LSST [8], yielding higher MOTA but lower IDF1. Nevertheless, CenterTrack [30] yields slightly higher accuracy on both metrics.

Similarly, on KITTI, our approach outperforms SORT [2], FAMNet [5], and mmMOT [29], but yields lower performance than CenterTrack [30].

**Qualitative Results.** We show qualitative results in Figure 5.

**Additional Experiments.** In the supplementary material, we report results for five additional experiments, where we compare MOTA on the MOT17 training set when various experimental parameters are changed, including detector accuracy, unlabeled video corpus size, and the training example sequence length $n$.

## 6 Conclusion

In this paper, we have shown that a robust, fully unsupervised multi-object tracker can be trained through a novel self-supervisory learning signal, cross-input consistency, that enforces consistency in the tracking outputs across different input variations of one video sequence. Despite training only on unlabeled video, our approach outperforms four supervised trackers published in the last 1–2 years (Tracktor++ [1], FAMNet [5], GSM [18], and mmMOT [29]), which train on expensive video-level bounding box and track annotations.

**Social Impact.** By enabling a robust multi-object tracker to be trained given only unlabeled video, our work promises to greatly reduce the effort for users to apply multi-object tracking on new datasets without sacrificing accuracy. Thus, we believe that our novel self-supervised MOT method can open up new video analytics tasks that were previously too costly. This impact may be positive or negative depending on the nature of these tasks — however, in general, we believe that tasks with greater potential for negative impact such as surveillance and pedestrian tracking would not benefit from the reduction in annotation cost associated with our method.

**Funding Transparency Statement.** This research was supported in part by the Qatar Computing Research Institute (QCRI).

---

[1]These results are taken from `https://motchallenge.net/results/MOT17/`, where our method is denoted UNS20regress. Baselines are denoted SORT17, IOU17, Tracktor++, MHT_bLSTM, FAMNet, LSST17, GSM_Tracktor, and CTTrackPub.

[2]These results are taken from `http://www.cvlibs.net/datasets/kitti/eval_tracking.php`.

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
