# Self-Supervised Multi-Object Tracking with Cross-Input Consistency (Supplementary Material)

## Favyen Bastani, Songtao He, Sam Madden

In this appendix, we detail occlusion-based hiding, and also include results for five additional experiments:

1. Varying Detector Performance (Training): MOTA on MOT17 when using detectors of varying performance during self-supervised training of our tracker model.

2. Varying Detector Performance (Inference): MOTA when using detectors of varying performance during inference, with the same tracker model parameters.

3. Varying Unlabeled Video Dataset Size: MOTA when self-supervised learning is conducted on video datasets of varying size.

4. Varying Sequence Length: adjusting the length of video sequences that are sampled on each training step.

5. Randomly Initialized Model: comparing performance of our approach against a randomly initialized model.

## 1 Occlusion-based Hiding

At a high level, occlusion-based hiding produces the variations $A(D)$ and $B(D)$ by simulating random occlusion incidents where all detections in occluded frames are eliminated from the input, i.e., if $I_k$ is occluded for $A(D)$, then $D_k^A$ is empty. We only occlude intermediate frames (i.e., a frame $I_k$ is only considered for occlusion if $0 < k < n$) so that the transition matrices still compare detections in $I_0$ with those in $I_n$. When processing an occluded $I_k$, the tracker is forced to match all tracks to the absent column in that frame, and re-localize the tracks after the occlusion.

We first introduce two schemes that do not work in isolation, and then show that we can combine these schemes to produce input variations that result in effective training.

**Only-Occlusion.** For each training sequence $\langle I_0, \ldots, I_n \rangle$, Only-Occlusion randomly selects four indexes $0 < k_1 \leq k_2 < k_3 \leq k_4 < n$ to construct two disjoint frame subsequences $\langle I_{k_1}, \ldots, I_{k_2} \rangle$ and $\langle I_{k_3}, \ldots, I_{k_4} \rangle$. In $A(D)$, we occlude each frame $I_k$ such that $k_1 \leq k \leq k_2$, and in $B(D)$, we occlude $I_k$ if $k_3 \leq k \leq k_4$.

When training under Only-Occlusion, the tracker is forced to leverage track features computed through the RNN to relocalize tracks after each simulated occlusion ends. Learning to merely compare detection features across consecutive frames would yield low accuracy since features in occluded frames are not observed. Furthermore, because one

tracker observes the detections and the other tracker does not, the model must make similar tracking decisions when re-localizing across occluded frames as it does when observing detections in each frame.

However, in practice, Only-Occlusion yields a model that simply memorizes detections in $I_0$, and computes $A^{(0,n)}$ and $B^{(0,n)}$ by comparing detections in $I_n$ against memorized detections. This strategy yields high consistency because it is unaffected by occluded intermediate frames. Thus, to make this scheme effective, we must prevent the propagation of features directly from $I_0$ to $I_n$.

**RNN Hand-off.** RNN Hand-off prevents simple memorization by cutting off the propagation of RNN features through the application of two separate RNN executions. We select two indexes $0 < k_5, k_6 < n$. Instead of computing $A^{(0,n)}$ directly, we first apply the tracker on the frame sequence $\langle I_0, \ldots, I_{k_5} \rangle$ to derive a transition matrix $A^{(0,k_5)}$ that matches detections in $I_0$ with detections in $I_{k_5}$. We then independently apply the tracker on $\langle I_{k_5}, \ldots, I_n \rangle$ to derive another matrix $A^{(k_5,n)}$ that matches detections in $I_{k_5}$ with detections in $I_n$. We combine these matrices through the matrix product to compute $A^{(0,n)}$: we compute $A^{(0,n)} = A^{(0,k_5)} A^{(k_5,n)}$. Similarly, we compute $B^{(0,n)} = B^{(0,k_6)} B^{(k_6,n)}$.

This scheme forces the tracker to find the same unique detection in $I_{k_5}$ (and $I_{k_6}$) for two detections of the same object in $I_0$ and $I_n$ in order to maximize similarity between the matrix products. However, a tracker that learns to match tracks to detections by comparing only the detection features in consecutive frames will exhibit high similarity between $A^{(0,n)}$ and $B^{(0,n)}$ under this scheme.

**Combined Scheme.** Only-Occlusion and RNN Hand-off have opposite advantages and drawbacks. Thus, we combine these in our occlusion-based hiding scheme. We first select the two sequences for simulated occlusion, $\langle I_{k_1}, \ldots, I_{k_2} \rangle$ and $\langle I_{k_3}, \ldots, I_{k_4} \rangle$. Then, we randomly pick $k_5$ and $k_6$ such that $k_3 \leq k_5 \leq k_4$ and $k_1 \leq k_6 \leq k_2$, i.e., the hand-off for one tracker occurs when the other tracker observes a simulated occlusion.

Under this scheme, neither memorizing features in $I_0$ nor comparing detections solely in a pairwise frame-by-frame manner is an effective tracking strategy. Instead, the tracker must learn to leverage RNN features for re-localizing across simulated occlusion, while still ensuring the tracking deci-

sions reflect intermediate outputs.

## 2 Varying Detector Performance (Training)

First, we consider the impact of the object detection model that we employ during self-supervised training on the robustness of the resulting tracker model. We do not vary the detector during inference – instead, we always use the same MOT17 SDP detector. We expect that using a detector model that most closely reflects the detections that will be seen during inference will maximize MOTA; however, the parameters for the MOT17 detectors are not available.

To vary detector performance, during tracker training, we vary the input resolution for a Mask R-CNN model trained on COCO from 1024x576 to 448x256. At each resolution, we measure the mAP score each detector achieves over the MOT17 training set frames to validate that we are testing a substantial range of detector accuracy levels. We train our tracker model under visual-spatial hiding using each of the detector resolutions. Finally, we compute the MOTA when applying each trained model on the MOT17 training set, using the SDP detections included in the MOT17 dataset.

| Resolution | Detector mAP | Visual-Spatial MOTA |
|---|---|---|
| 1024x576 | 0.32 | 60.2% |
| 832x448 | 0.29 | 59.5% |
| 640x360 | 0.26 | 59.7% |
| 448x256 | 0.18 | 59.3% |

The table above shows the results. The detector provides higher accuracy at higher resolutions. Thus, the results suggest that there is a weak correlation between detector performance and resulting tracker accuracy. We hypothesize that this is in large part because the higher accuracy detections also correspond more closely with the MOT17 SDP detector.

## 3 Varying Detector Performance (Inference)

We also compare the performance of our tracker model trained under visual-spatial hiding when varying detector performance during inference, but keeping constant the detector used for self-supervised training. To do so, we simply report the MOTA achieved on the MOT17 training set under each of the object detectors included in the MOT17 dataset; ordered from lowest-accuracy to highest-accuracy, these are Deformable Parts Model (DPM), Faster R-CNN (FRCNN), and Scale-Dependent Pooling (SDP).

| | DPM | FRCNN | SDP |
|---|---|---|---|
| MOTA | 45.2 | 46.1 | 48.0 |

MOTA increases with detector accuracy.

## 4 Varying Unlabeled Video Dataset Size

We now consider the impact of the amount of unlabeled video (which we use during self-supervised training of the tracker model) on the robustness of the resulting tracker model. Note that unlabeled video can be cheaply obtained since no manual annotation is required to collect it. We vary the amount of unlabeled video by using 100%, 25%, 15%, and 5% of the PathTrack corpus (which totals 2.9 hours of video); we do not use the labels in PathTrack. We then compute the MOTA

when applying each model, trained under visual-spatial hiding, on the MOT17 training set.

| Unlabeled Video Percentage | MOTA |
|---|---|
| 100% | 59.2% |
| 25% | 58.3% |
| 15% | 57.3% |
| 5% | 56.3% |

The tracker performance rapidly deteriorates as the amount of unlabeled video is reduced. At 5% of the PathTrack corpus (9 minutes of video), the performance of our tracker model is similar to the performance of SORT, which only uses spatial features (bounding box coordinates). This suggests that, when training with only 9 minutes of video, our method is able to learn to use spatial cues for tracking objects, but does not have sufficient training data to learn to leverage visual cues.

## 5 Varying Sequence Length $n$

Below, we report MOTA on the MOT17 training set of a model trained under visual-spatial hiding using varying sequence lengths. We also report the accuracy when the sequence length $n$ is randomly sampled from a set of multiple options on each training example.

| Sequence Length(s) | MOTA |
|---|---|
| 2 | 62.2% |
| 4 | 60.2% |
| 8 | 62.1% |
| 16 | 62.0% |
| 32 | 61.5% |
| 4, 8, 16, 32 | 62.1% |

Tracker performance is not very sensitive to the sequence length.

## 6 Randomly Initialized Model

To highlight the degree to which self-supervised learning improves performance over a randomly initialized model, we compare the performance of our method on the MOT17 training set against such a baseline. In the baseline, since a randomly initialized matching network will not effectively compare image and spatial features, we opt to eliminate the matching network and replace it with an L2 distance function between bounding box coordinates or extracted image features. It achieves -76.4% MOTA (negative MOTA) and 2.8% IDF1, suggesting that random initialization is not at all effective, and that our cross-input consistency approach elevates performance.