# OpenReview forum: "Self-Supervised Multi-Object Tracking with Cross-input Consistency"
_NeurIPS.cc/2021/Conference — NeurIPS 2021 Poster_

### Official Review · Reviewer_rjLZ · 2021-07-15

**Rating:** 5
**Confidence:** 3

**Summary:**

This paper proposes a self-supervised learning procedure for training a robust MOT model given only unlabeled video. This unsupervised method outperforms several recent supervised methods.

**Limitations And Societal Impact:**

1. The two input is the objects detected in the first and last frames of the sequence. What if the object disappears or is occluded before the last frame？
2. The written need be improved and the figures should be more clear to explain the method. The related work should be described by different classes. It is best not to put Figure 1 and Figure 2 in the related work. In Figure 1, I suggest using real pictures instead of synthetic pictures to display the cross-input consistency. Input A and Input B is the different input while using the same image here.
3. The performance of Visual-Spatial is much better than SORT and IOU. But these two methods is too old (published in 2016 and 2019). Several more recent methods (unsupervised/semi supervised/supervised) should be added and discussed, such as Chained-Tracker [1] and QDTrack [2].
4. This paper focues on the training process, which is lack of the detailed description of the test process. Besides, the time-consuming (fps) of each module should be analyzed and compared with other methods.

[1] Chained-Tracker: Chaining Paired Attentive Regression Results for End-to-End Joint Multiple-Object Detection and Tracking. ECCV, 2020.
[2] Quasi-Dense Similarity Learning for Multiple Object Tracking. CVPR, 2021.

**Main Review:**

Originality: The submission proposes a new self-supervised architecture for multi-object tracking, which is novel.

Quality: The submission is technically sound. The author submits the code, which is convenient to reproduce.

Clarity: The written is not so clear.

Significance: The performance improvement is general compared to the existing methods.

**Time Spent Reviewing:**

2 hours

---

> ### Author Response · Authors · 2021-08-09
> **Response**
>
> We thank the reviewer for their comments. Below, we address concerns that were raised.
>
> > The two input is the objects detected in the first and last frames of the sequence. What if the object disappears or is occluded before the last frame？
>
> In Section 3.2, we discuss the selection of the sequence length n: specifically, it should be set so that objects undergo non-negligible motion over the sequence, but most objects that appear in the first frame are still visible in the last frame. If the model is trained in this way, then on training sequences where it encounters objects that are not visible in the last frame, the model will learn to assign the probability mass of those objects to the ‘absent’ column in the transition matrix (indicating the object was not present in that frame), since this maximizes the matrix similarity score in expectation.
>
> > The written need be improved and the figures should be more clear to explain the method. The related work should be described by different classes. It is best not to put Figure 1 and Figure 2 in the related work. In Figure 1, I suggest using real pictures instead of synthetic pictures to display the cross-input consistency. Input A and Input B is the different input while using the same image here.
>
> We thank the reviewer for the comments on clarity. We will update the paper to address these comments, including improving Figure 1.
>
> > The performance of Visual-Spatial is much better than SORT and IOU. But these two methods is too old (published in 2016 and 2019). Several more recent methods (unsupervised/semi supervised/supervised) should be added and discussed, such as Chained-Tracker [1] and QDTrack [2].
>
> Chained-Tracker and QDTrack are supervised methods that train their object association components on video-level annotations. We compare against other similarly recent supervised methods, including GSM (IJCAI 2020) and CenterTrack (ECCV 2020). QDTrack was not published until after our submission. Nevertheless, we will consider adding Chained-Tracker and QDTrack as additional baselines in our evaluation.
>
> > This paper focues on the training process, which is lack of the detailed description of the test process. Besides, the time-consuming (fps) of each module should be analyzed and compared with other methods.
>
> We will improve the clarity of the inference procedure description. The runtime is dominated by the object detector inference time, since the detector is the only component that processes the entire video frame. The overall runtime of our method is available on the MOT17 benchmark website; we will incorporate it into the supplementary material. We will consider investigating the runtime of each module as well.

---

### Official Review · Reviewer_4Dyg · 2021-07-15

**Rating:** 4
**Confidence:** 4

**Summary:**

This paper proposes a self-supervised learning scheme for training Multi-Object Tracking (MOT) models using only unlabeled video. The proposed scheme constructs two distinct inputs by hiding different information about the sequence in each input and trains the model to produce consistent tracks across the two inputs. The proposed algorithm is evaluated on the MOT17 and KITTI datasets with comparisons against several supervised methods.

**Limitations And Societal Impact:**

Yes

**Main Review:**

Novelty and Contributions:
The idea of using the cross-input consistency, which trains the model to produce consistent tracks on two inputs with different hiding information, to train MOT models on unlabeled video is novel.
In addition, the authors provide two input-hiding schemes (Occlusion-based hiding and visual-spatial hiding) for generating input variations.
Exploiting self-supervised learning signals to train MOT models on videos without annotations is meaningful.

However, I have the following concerns.
-	Limitations. Although the proposed method can be trained on unlabeled videos, it requires an object detector for the object category of interest, to provide object bounding boxes. The proposed algorithm only exploits the self-supervised learning scheme to learn the data association module for matching tracks and detections. It can be only applied in tracking-by-detection MOT methods, which cannot be trained end-to-end. In addition, since the detection part also belongs to the MOT framework, it may be inappropriate to call the proposed MOT tracker a fully unsupervised one.
-	Line 178 writes ‘in general, we find that setting n to one-half of the average time that objects linger in the camera frame works well.’. But how to find the average time on large-scale training videos is not given. This assumption also limits the selection of training data and the diversity of training samples.

Experimental results:
The experimental results are good. However, I have the following concerns.
-	Line 275 writes ‘Like our approach, SORT and V-IOU require an object 276 detector,’. Since the authors do not state clearly the test settings on the MOT17 dataset, I am wondering whether the public detections or the private detections (generated by the yolov5 detector) are used?
-	For the ablation study, adding a baseline that directly performs tracking without training may be helpful, since the features of the same object are usually similar. The comparison against this baseline will better show how much the unsupervised learning scheme contributes to the final performance.

Clarity:
This paper is not well organized.
It takes more than 1 page (from Line 216 to Line 259) to describe the Occlusion-based hiding and uses a 1/4 page to introduce the visual-spatial hiding. The results (in Table 1) show that the later scheme achieves better performance. Line 310 writes that ‘when training under occlusion based hiding, the model is unable to learn to re-localize objects over simulated occlusions since the simulated occlusion must then also be short.’, which indicates that the former occlusion-based hiding scheme does not work well.
Finally, the authors use the visual-spatial (later) scheme to compare with other methods.
This is really confusing. If the authors know the occlusion-based hiding scheme does not work well, then why spend more than 1 page to describe it. In addition, some details about the visual-spatial scheme are missing.





**Time Spent Reviewing:**

4

---

> ### Author Response · Authors · 2021-08-09
> **Response**
>
> We thank the reviewer for their comments. Below, we address concerns that were raised.
>
> > Limitations. Although the proposed method can be trained on unlabeled videos, it requires an object detector for the object category of interest, to provide object bounding boxes. The proposed algorithm only exploits the self-supervised learning scheme to learn the data association module for matching tracks and detections. It can be only applied in tracking-by-detection MOT methods, which cannot be trained end-to-end. In addition, since the detection part also belongs to the MOT framework, it may be inappropriate to call the proposed MOT tracker a fully unsupervised one.
>
> We agree that our method requires an object detector trained on image-level annotations. Our contribution is to develop a self-supervised tracking approach that requires no video-level annotations, which are substantially more expensive to label than image-level annotations. Our method is comparable to other unsupervised multi-object tracking methods such as SORT and IOU that also depend on an object detector.
>
> > Line 178 writes ‘in general, we find that setting n to one-half of the average time that objects linger in the camera frame works well.’. But how to find the average time on large-scale training videos is not given. This assumption also limits the selection of training data and the diversity of training samples.
>
> The n parameter should be set so that tracks visible in the first frame undergo non-negligible motion during the video sequence but do not all leave the camera window before the last frame. The one-half average track duration value is a rough setting that achieves this. Our method is robust to minor changes in the parameter. Thus, one simple and cheap way to select n on large-scale training videos would be to sample 10 or so timestamps, label the duration of the next object that enters the camera window following each of those timestamps, and set n to half the average duration. This procedure can be completed in just one or two minutes. We will further explore the sensitivity of our method to different settings for n in the supplementary material.
>
> > Line 275 writes ‘Like our approach, SORT and V-IOU require an object 276 detector,’. Since the authors do not state clearly the test settings on the MOT17 dataset, I am wondering whether the public detections or the private detections (generated by the yolov5 detector) are used?
>
> We use the public detections for MOT17. The accuracy numbers in Table 2 are taken from the benchmark website, which computes the average across the three detectors provided by MOT17. We will clarify this in the paper. We use YOLOv5 on KITTI since KITTI does not provide detections.
>
> > For the ablation study, adding a baseline that directly performs tracking without training may be helpful, since the features of the same object are usually similar. The comparison against this baseline will better show how much the unsupervised learning scheme contributes to the final performance.
>
> We agree that adding such a baseline could be insightful. However, the matching network parameters still need to be learned in order to produce meaningful scores in the transition matrix. We will consider adding a baseline like this where the matching network is replaced by a fixed function that computes a score from the L1 or L2 distance between the track-level and detection-level features.
>
> > Clarity
>
> We thank the reviewer for the comments on clarity, and we will update the paper to address these comments. Regarding the input hiding schemes, our main contribution is our cross-input consistency approach, which can operate with various input hiding schemes. We describe both schemes to emphasize the generality of our approach, and to compare the performance of different input hiding schemes (Table 1). Note that, while visual-spatial hiding outperforms occlusion-based hiding, occlusion-based hiding still outperforms several other baselines such as SORT. Nevertheless, we agree that, since visual-spatial hiding outperforms occlusion-based hiding on MOT17 and KITTI, it should be allocated more space in the paper. Thus, we will reduce the space spent on occlusion-based hiding, and use it to describe visual-spatial hiding and other components of our approach in more detail. We will also add experiments to validate the effectiveness of occlusion-based hiding compared to several baselines.

---

> > ### Comment · Reviewer_4Dyg · 2021-09-10
> > **Response**
> >
> > After reading all the comments and responses, I find several crucial concerns have not been addressed.
> >
> > The authors agree with most of my concerns, such as
> > 1) missing discussion about choice/effect of hyperparameters (n);
> > 2) section 4.1 is included in the method section but not used in the final method;
> > 3) still requires a trained detector to work;
> > 4) missing ablation study (missing baseline performing tracking without training).
> >
> > The first two concerns are also recognized by R1.
> > The authors said they will improve these, however, there is too much to be improved for this paper.  Thus, I keep my initial rating of 'Ok but not good enough' unchanged.

---

### Official Review · Reviewer_fDwk · 2021-07-17

**Rating:** 7
**Confidence:** 5

**Summary:**

This paper presents a novel framework to train the multi-object tracker in an unsupervised scheme. The framework utilizes the cross-input consistency assuming that the two clips manipulated by one video must have the same tracking trajectory. The two manipulated clips are randomly generated by one of the occlusion-based hiding and visual-spatial hiding. By using the baseline of the RNN-based tracker, the proposed framework shows comparable performance even with the supervised MOT.

**Limitations And Societal Impact:**

Please read the main review.

**Main Review:**

The framework seems novel and its effectiveness is also impressive. In addition, a detailed description is given as well sufficiently to reproduce the overall framework. The definition and notations were clear to understand the implementation.

I have several questions and comments.

- The relation between the number of data and the accuracy
  Since the proposed framework can utilize infinite data to train the tracking model, the performance can be improved when more variety of videos are trained. I hope to know when the performance improvement converges and why.

- Full clip instead of the manipulated clip?
  The proposed framework utilizes two manipulated clips. However, when one of the two clips is replaced by a full video clip that suffers from no manipulation, the training would be more stable. Since the tracking performance in the full video clip would be the upper bound of the performance in the manipulated video clip, we may be able to acquire the increased performance. What is the reason why both two video clips should be manipulated?

- Sensitivity to the noisy detector
  This framework basically assumes that the detection box is reliable. Then, what would be happened when the noisy detection is given in the self-training phase? Since all the experimental results utilized one specific detector, we cannot find the sensitivity to the noisy detection. I recommend the authors show some results with the different detectors or the noisy detection boxes.

**Time Spent Reviewing:**

2 hours

---

> ### Author Response · Authors · 2021-08-09
> **Response**
>
> We thank the reviewer for their comments. Below, we address concerns that were raised.
>
> > The relation between the number of data and the accuracy
>
> We do show some experiments regarding the amount of data in the supplementary material. At 5% of the unlabeled training corpus, our method yields 56.3% MOTA, but with 100%, it yields 59.2% MOTA.
>
> > Full clip instead of the manipulated clip? The proposed framework utilizes two manipulated clips. However, when one of the two clips is replaced by a full video clip that suffers from no manipulation, the training would be more stable. Since the tracking performance in the full video clip would be the upper bound of the performance in the manipulated video clip, we may be able to acquire the increased performance. What is the reason why both two video clips should be manipulated?
>
> We agree that the input hiding schemes could in principle provide the full video clip as one of the inputs. For visual-spatial hiding, both clips must be manipulated since otherwise the model could learn to discard either the visual or spatial features when provided with the full clip to obtain the other input, and thereby achieve high similarity. For occlusion-based hiding, it should be possible for one input to be the full clip, but we have not explored this.
>
> > Sensitivity to the noisy detector
>
> We do show some experiments regarding varying detector performance in the supplementary material. In these experiments, we vary the detector used during training, but keep the detector used during inference fixed. To obtain detectors with different accuracy levels, we train the detectors to input video at varying resolutions. At 448x256 detector resolution, our approach yields 59.3% MOTA, while at 1024x576 detector resolution, our approach yields 60.2% MOTA. We will expand this experiment to include the mAP accuracy of the detector at each resolution.
>
> We have also evaluated our method with varying detector performance during inference, with the detector used during training fixed, since the MOT17 test set provides detections computed by three approaches (Deformable Parts Model, Scale-Dependent Pooling, and Faster R-CNN) that exhibit very different accuracy levels. We only report the average performance in the paper, but these results are available on the benchmark website. We will incorporate the results into the supplementary material.

---

### Official Review · Reviewer_thph · 2021-07-17

**Rating:** 7
**Confidence:** 4

**Summary:**

This paper presents a method for self-supervised multi-object tracking in videos. The authors propose a cross-input consistency loss to train their RNN-based tracking method to form correct associations between detected objects in the various video frames and also to relocate them when tracks are lost. The core idea of the cross-input consistency loss is to present two different (altered) versions of the same video sequence to the tracking algorithm and to ensure that despite differences in the exact sequences the algorithm is able to identically associate the detections in the first and last frames of the two sequences. The authors present various methods to alter the sequences so as to avoid degenerate solutions from being learned. Their method shows significant improvements over the existing unsupervised tracking algorithms and are competitive with several existing SoTA supervised approaches on two benchmark datasets -- MOTA17 and KITTI.

**Limitations And Societal Impact:**

The authors have addressed the (positive and negative) societal impact of their work.

They should include more discussion of the $n$ parameter used in their algorithm and how it should be selected in real-world settings without video annotations.

They should remove the discussion of the occlusion-hiding section from the proposed method as it is never used in their final method and is mis-leading.

They should also remove speculative and unsupported claims about [24] and [25] not being able to work with RNNs.



**Main Review:**

Originality: The paper does not propose a completely novel task, but rather the first learning-based algorithm for the task of self-supervised video multi-object tracking. Little research has been done for this task before so in that regard this work represents the first dedicated and systematic attempt at trying to solve this problem and hence is sufficiently novel.

Quality: Overall the proposed approach is well-described and sufficiently motivated. However, I have some major concerns about some parts of the proposed approach. I describe them below.

1. The biggest concern that I have is that from line 316 and Table 2 it is clear that in their final algorithm the authors employ the "Visual-Spatial Hiding" only. Why then is the description of occlusion-hiding (section 4.1) even included in the method section paper since it is never used in the final method? This largely undermines a significant portion of the proposed method and I feel that to some extent the authors have mis-represented their proposed method.

2. The authors make the speculative claim several times that the existing colorization [24] and cycle-based [25] techniques for learning with RNNs "would not work" -- in the abstract and lines 38-39. However, this claim is not supported by the experiments shown in the paper. I believe that the "Combined Occlusion Scheme" proposed by the authors along with a forward and backward cycle loss from [25] could be applied to train the RNN while ensuring that it does not just memorize the transitions between frame 0 and $n$ only and to force it to learn the transitions in between in the sequence as well. Anyhow, the authors should either verify this claim with experimental evidence or remove it entirely from the paper.

3. In lines 170-179 the authors describe that the success of their algorithm depends on the assumption that most objects presents in frame $0$ should also be present in frame $n$. They further mention that this critical parameter $n$ can be set as "one-half of the average time that objects linger in the camera frame". However, this requires knowledge of labeled tracks in videos. Can the authors comment on this limitation of their algorithm? How critical is this parameter to the overall success of their unsupervised learning algorithm? Lines 308-312 seem to indicate that the proposed algorithm is indeed quite sensitive to this parameter.

Have the authors systematically analyzed its sensitively for a particular dataset and across datasets? Related to this, have the authors studied if the availability of longer tracks in general results in better overall performance for their algorithm?

4. Have the authors analysed if domain differences between the labeled and unlabelled datasets affect the accuracy of their approach and ways to mitigate these?

Clarity: The material presented is mostly clear. However, some parts of the method are not explained clearly:

1. "When processing an occluded $I_k$, the tracker is forced to match all tracks to the absent column in that frame, and re-localize the tracks after the occlusion." Is this something that is enforced by design or something that emerges automatically in the transition matrix probabilities during training? How does the subsequent re-localization happen, when all tracks get assigned to a single "absent" detection? Are all detections in a subsequence of frames hidden or just a few of them? This part is not very clear and it would be helpful to explain this more clearly by illustrating transition matrices and how they change in this case.

2. line 295: When $n$ is set to 4 during training, there are only 3 frames in between the first and last frame. How to the authors set $k1$, $k2$, $k3$, $k4$, $k5$ and $k6$ in this case?

3. In the tables it would be good to add (up/down) arrows to indicate the direction of improvement of the metric.

4. Why are the results of the proposed algorithm (Visual-spatial) different in Tables 1 and Table 2?

5. For the experiments on MOT17, how were the frame-wise detections obtained?

Significance: Obtaining reliable video-level annotations is a real practical problem while deploying video-based tracking algorithms, which limits their applicability currently. This paper addresses this existing open problem, which has much practical importance. The results of the proposed method also present a significant improvement over the existing unsupervised methods and hence this work advances the research in this field. However, I have several concerns about the presented method and how it was implemented, which I would like to hear more clarification about from the authors.

Post rebuttal: The authors have adequately addressed my concerns. The paper addresses an important problem. I recommend acceptance and have raised my score.

**Time Spent Reviewing:**

6

---

> ### Author Response · Authors · 2021-08-09
> **Response**
>
> We thank the reviewer for their comments. Below, we address concerns that were raised.
>
> > The biggest concern that I have is that from line 316 and Table 2 it is clear that in their final algorithm the authors employ the "Visual-Spatial Hiding" only. Why then is the description of occlusion-hiding (section 4.1) even included in the method section paper since it is never used in the final method?
>
> Our main contribution is our cross-input consistency approach, which can operate with various input hiding schemes. We describe both schemes to emphasize the generality of our approach, and to compare the performance of different input hiding schemes (Table 1). Note that, while visual-spatial hiding outperforms occlusion-based hiding, occlusion-based hiding still outperforms several other baselines such as SORT. Nevertheless, we agree that, since visual-spatial hiding outperforms occlusion-based hiding on MOT17 and KITTI, it should be allocated more space in the paper. Thus, we will reduce the space spent on occlusion-based hiding, and use it to describe visual-spatial hiding and other components of our approach in more detail. We will also add experiments to validate the effectiveness of occlusion-based hiding compared to several baselines.
>
> > The authors make the speculative claim several times that the existing colorization [24] and cycle-based [25] techniques for learning with RNNs "would not work" -- in the abstract and lines 38-39. However, this claim is not supported by the experiments shown in the paper.
>
> Our intention was to argue that the most simple extension of [25] for RNN models, where we incorporate hidden features into the single-object tracking model but do not modify anything else, would be ineffective. We agree that, for example, combining it with a method akin to occlusion-based hiding could work. We will clarify our claim in the paper to ensure its very limited scope is explicit, and eliminate the claim for [24].
>
> > In lines 170-179 the authors describe that the success of their algorithm depends on the assumption that most objects present in frame should also be present in frame . They further mention that this critical parameter can be set as "one-half of the average time that objects linger in the camera frame". However, this requires knowledge of labeled tracks in videos.
>
> We will expand on both the sensitivity of our method to this parameter, and its selection. On sensitivity, as we discuss in Section 3.2, we find that as long as it is set in a range where tracks undergo non-negligible motion but still smaller than the average track duration, our method performs similarly. We will add experiments to the appendix to explore this further, where we train our model with varying settings for n. On selection, one simple and cheap sampling-based way to approximate the average track duration would be to sample 10 or so timestamps in the video, label the duration of the next object that enters the camera window following each of those timestamps, and compute the average over those tracks; this can be done in just one or two minutes, which is substantially faster than labeling sufficient video-level bounding box annotations for training fully supervised methods.
>
> > Have the authors analysed if domain differences between the labeled and unlabelled datasets affect the accuracy of their approach and ways to mitigate these?
>
> We did assume that minimizing the domain differences between the unlabeled training data and the test data would be important to achieving good performance. Thus, for example, we chose to use PathTrack data (without labels) and walking tours that are similar to the MOT17 sequences. One advantage of our approach is that, since video is generally captured in large quantities, there will typically be tens or even hundreds of hours of video in the application domain available for training; furthermore, as the domain shifts, our approach can simply be periodically re-trained without needing to collect new video-level labels. We will expand on this point in the paper. We agree that data augmentation approaches may enable training on data with substantial domain differences, but we have not explored this.
>
> > Clarity
>
> We thank the reviewer for the comments on clarity. We will update the paper to address these comments. Regarding differences between Tables 1 and 2, Table 1 shows performance on the training set where we can run all ablations while Table 2 shows performance on the test set where we can only submit the best approach to the challenge website for evaluation. Regarding frame-wise detections for MOT17, we use the public DPM, FRCNN, and SDP detections provided by the benchmark.

---

> > ### Comment · Reviewer_thph · 2021-09-02
> > **Response to Authors**
> >
> > I thank the authors for their carefully written response to all my concern.
> >
> > The authors have addressed all my concerns adequately. I encourage the authors to incorporate the suggested changes in the final paper.
> >
> > The paper proposes an interesting approach for temporal association of bounding boxes, which even with the requirement of individual frame-level bounding box detectors, in of itself an important non-trivial problem requiring significant labeling effort.
> >
> > Hence, I have raised my final score and I recommend accepting this paper.

---

### Decision · Program_Chairs · 2021-09-27

**Decision:**

Accept (Poster)

**Comment:**

The paper initially received mixed reviews, two for (6 & 7) and two against (4 & 5).  The reviewers appreciated the motivation and novel self-supervised learning method proposed for MOT, as well as the superior results against previous unsupervised methods, as well as many supervised methods.  The main concerns from the reviewers were:

1. missing discussion about choice/effect of hyperparameters (n).
2. "occlusion-based hiding" is proposed (and a lot of space is dedicated to describing it), but performs worse than the proposed "visual-spatial hiding".
3. some claims not supported
4. still requires a trained detector to work, which can be considered as not fully unsupervised MOT.
5. missing ablation studies and baselines.
6. use more recent comparison methods for MOT, Chained-Tracker and QDTrack (published after NeurIPS deadline).

The author provided a response, and attempted to address these points through further explanations and promises to rewrite/expand parts of the paper. Some ablation studies were already presented in the supplemental. After the discussion one positive reviewer raised from 6 to 7, while the other positive reviewer maintained 7. One negative reviewer was concerned that a large amount of changes are required to address the review comments, and thus maintained rating 4.  The other negative reviewer did not participate in the discussion. The AC checked the response to this reviewer's comments, and thinks that the responses to their comments were satisfactory.  In the end the AC has a positive impression on the paper, and thus recommends acceptance, under the condition that the presentation is improved.

The authors should revise the paper according to the reviews and discussion, in particular: reduce the discussion on occlusion-based hiding, increase discussion on visual-spatial hiding, include the ablation studies from the supplemental, add baselines (tracking w/o training), expand discussion about parameter n.